# Social presence and dynamics of group communication: An analysis of a health professionals WhatsApp group chats

Chidozie E. Mbada[1]*, Oluwatosin O. Jeje[2], Micheal Akande[2], Kikelomo A. Mbada[3], Clara Fatoye[1,4], David Olakorede[2], Olusola Awoniyi[2], Udoka A. C. Okafor[5], Olatomiwa Falade[6], Francis Fatoye[1]

1 Department of Health Professions, Faculty of Health and Education, Manchester Metropolitan University, Manchester, United Kingdom, 2 Department of Medical Rehabilitation, College of Health Sciences, Obafemi Awolowo University, Ile-Ife, Nigeria, 3 School of Digital Education, Faculty of Learning and Teaching, Arden University, Manchester, United Kingdom, 4 Faculty of Health and Care Professions, University Campus Oldham (UCO), University Way, Oldham, United Kingdom, 5 Department of Physiotherapy, Faculty of Clinical Sciences, University of Lagos, Ikeja, Nigeria, 6 Royal Oldham Hospital, Oldham, Greater Manchester, United Kingdom

* c.mbada@mmu.ac.uk, doziembada@yahoo.com

**Data Availability Statement:** All relevant data are within the paper and its Supporting Information files.

## Abstract

WhatsApp has become a medium of communication with the potential of promoting collaborative environment with peers, patients and general population. Till date, no analysis of professional WhatsApp groups' activities exists in physiotherapy. The official WhatsApp group chats of the Association of Clinical and Academic Physiotherapists of Nigeria (ACAPN) was analyzed. A total of 20760 chats were gleaned from July 2020 to June 2021. Videos, audios and emoticons were excluded in the analysis. Administrative permission to conduct was obtained from ACAPN leadership. Two consenting physiotherapists who had never deleted their ACAPN group chats shared and exported all chats to a Gmail. The social presence theory for group communication was used as the framework of analysis. Thematic content analysis was used to analyze qualitative data. Descriptive statistics of frequency and percentages were used to summarize data. Based on social presence theory classifications, interactive messages (64.5%) followed by cohesive messages (30%) were predominant. Members used the platform more for expression of emotions affectively (100%), referring explicitly to others' messages interactively (56.6%) and for greetings (phatic and salutation) cohesively (61.8%). Qualitative themes indicate that all three categories of social presence theory communications were present sufficiently with interactive category being the most common, as members used the WhatsApp platform to interact, construct and share knowledge. Group WhatsApp platform is a veritable means of communication and an indicator of level of social presence among Nigerian physiotherapists. Communication among Nigerian physiotherapists is mostly interactive, then cohesive and affective in terms of dynamics.

**Funding:** The author(s) received no specific funding for this work.

**Competing interests:** The author(s) received no specific funding for this work.

## Introduction

The advent of information technology and the current rise of mobile phones and mobile apps has brought about significant evolution in healthcare service delivery, education and learning methods, networking/collaboration, professionals and peer-to-peer support facilitation [1]. Innovations in the area of social media mobile tools such as Instagram and WhatsApp have brought about notable paradigm shifts to social networking services [2], as well as, serving as a worthwhile platforms for sharing information, discussing professional matters and social messages in form of texts, photos, audios and videos [1]. Social networking platforms such as WhatsApp are finding important niche among professional groups such as healthcare service providers [1]. WhatsApp which is the third most used social media app in the world, is house to several groups and professional organizations who used the platform as a means to communicate and encourage social and mobile forms of learning and meetings [3]. Nardo *et al.* [4] in a study on a 'WhatsApp Surgery Group' found that WhatsApp platform facilitated communication and enhances learning. In another study by Dorwal *et al.* [5] on a "WhatsApp laboratory groups", significant improvements in communication were seen in sharing photographic evidence, information about accidents, critical alerts, academic activities and directives. Conversely, Dorwal *et al.* [5] documents that Group WhatsApp have chats that are considered unwelcome and disturbing in routine workflows, however, the benefits of keeping Group WhatsApp platforms outweighed the minor hassles associated with the use. Also, Johnston *et al.* [6] analyzed WhatsApp communication method among emergency surgery teams in a London hospital and found that the app helped to "flatten hierarchy" amongst students, residents and experienced consultants, enabling them to actively contribute to discussions without inhibition.

In another study on WhatsApp as an intradepartmental communication tool, Khanna *et al.* [7] found that WhatsApp can bring about an improvement in patient-related awareness, communication and handovers among orthopedic residents. Among students, Robinson *et al.* [8] found that using WhatsApp helped to develop social presence among first-year radiography undergraduates, while Willemse [9] found that using WhatsApp improved undergraduate nurses' primary healthcare education. Similarly, other researchers found that WhatsApp can improve peer engagement and increase learners' participation, attitudes and achievement levels [10]. Therefore, the application is very significant in mastering spontaneous interaction in mobile learning and information exchange [11].

Wani and colleagues [12] utilized WhatsApp platform as a means communication in patient management and as a tool for academic endorsements among staff of plastic and reconstructive surgery. Other studies have found WhatsApp as an accessible, useable and efficacious platform for inter-and-intra professional communication among health care providers [13–15]. As an emerging area in academic research, the dynamics of group communication on social media, such as WhatsApp is yet to be sufficiently explored. Considering that WhatsApp's Group Chat feature users chat and share contents such as texts, videos, voice messages and images with up to 256 people at once, and can send broadcast messages without having to select them each time [16–18] thus, it is a veritable platform for data.

Nigeria's smartphone penetration is projected to grow to 64.9 percent in 2025, as in 2020, the number of mobile internet users in Nigeria amounted to over 85.26 million (Nigeria mobile internet users, 2015–2025 [19]. This national estimate is expected to be trickledown to professional group usage of smartphone and mobile apps such as WhatsApp. The huge statistics on mobile penetration and WhatsApp usage, present an opportunity to contribute to this recent area of WhatsApp studies. In Nigeria, there seems to be an apparent dearth of scholarly reports on WhatsApp group communication among any professional group. Characterization

of chats on professional groups WhatsApp platforms may give better understanding of the usefulness of such groups viz-a-viz the purpose for which they are meant to serve. Till date, no analysis of professional WhatsApp groups' activities exist in physiotherapy. In order to widen the academic study area of WhatsApp by analyzing the dynamics of group communication through this application, the study aimed to profile the level of social presence of the WhatsApp group of the Association of Clinical and Academic Physiotherapists of Nigeria (ACAPN) and to characterize the pattern of communication of ACAPN on Professional WhatsApp group, taking into account the social presence theory in group communication.

## Materials and methods

WhatsApp chats of a physiotherapy professional group (ACAPN) were used in this study retrospective review. The official WhatsApp platform of ACAPN had 238 physiotherapists on the platform. Chats from all the physiotherapists were used, these constitute the sample, and each chat the unit of analysis. As such, all chats on the WhatsApp group within one year time frame from July 1, 2020 to June 30, 2021, were extracted for analysis. In order to ensure representativeness of chats, two consenting group administrators who had never deleted any chat message within the study period were approached for permission for harvesting of data from their mobile phone. On their permission, a copy of the group chat history was exported to the Gmail of one of the authors (OOJ) for harvesting from each of them. Data from both sources were converted to word document. Reproducibility analysis for the data obtained from the two group administrators chats was done by painting each administrator's chats a different colour before they were merged. Then, the chats were sorted for reproducibility. It was found that chats obtained from both administrators were all paired, confirming that none of the physiotherapists had deleted any chat within the time frame that was studied. Thereafter, duplicate chats were deleted. Personal identifying information were deleted from the chats (including names, initials, gender, and phone numbers) and were coded appropriately. The text file obtained from the mail was converted to word document and was safely passworded.

The Social Presence Theory was used as the framework to define the themes in this study. Social presence is an important factor that help to build and promote interaction and create sense of community [20–23]. It is defined as "the awareness of others in an interaction, combined with an appreciation of the interpersonal aspects of that interaction" [22, 23]. Rourke and colleagues [21] argued that social presence indicators remain:

Affective Responses–This indicator is essentially about emotional expression involving use of humor and self-disclosure. Humor, as a communication skill is a form of stimulation that aims to elicit the laughter reflex, by using words and expressions that are sudden, lively, comical, or absurd for their effect [24, 25]. Humor is considered as an efficient way to introduce or modulate communication [26]. Also, self-disclosure, which is the process of communication or sharing personal details about oneself with another [27, 28], has been acknowledged to enhances relationship quality to a greater degree during online communications [29].

Interactive Responses—This indicator is also referred to as 'Open Communication'. It is characterized by continuing thread of communication involving asking questions, making reference to group members messages, giving compliments and communicating one's appreciation.

Cohesive Responses–This indicator is also known as 'Group Cohesion', and its elements include vocatives (defined by addressing, referring or qualifying members by names), phatics/salutations (this refers to greetings, closures that serves purely social function) and referring to the group using inclusive pronouns [21].

The researchers made official application for permission to conduct this study to the group. No member of the group disapproved the request. Consequently, an official signed letter to conduct study was issued by the leadership of the group. Ethical approval for the study was obtained from the Health Research Ethics Committee of the Institute of Public Health, Obafemi Awolowo University, Ile-Ife, Nigeria (IPH/OAU/12/1750). Owing to the retrospective nature of this study, chats records were fully anonymized by OOJ and MA who glean data were not members of the group and are blinded to any personal identifiable information.

## Statistical analysis

Both quantitative and qualitative data synthesis was used in analysis of the chats. Qualitatively, chats were analyzed using thematic content analysis. The social presence indicators were used to draw the themes. Descriptive statistics of frequencies and percentages was used to profile charts according to message, and social presence indicators (affective–expression of emotions, use of humor and self-disclosure; interactive–quoting from others' messages, referring explicitly to others' messages, asking questions, complimenting and expressing appreciation, expressing agreement; and cohesive–vocatives, addresses or refers to the group using inclusive pronouns, phatic and salutations) [21]. The Atlas.ti software was used for the thematic content analysis.

## Results

### Quantitative presentation of findings

From Wednesday, 1 July 2020 to Wednesday, 30 June 2021, there are 364 days. Overall, a total of 20760 texts were harvested and analyzed. Thus, approximately 611 chats per day, and 3 chats per person/day. In this study, affective category made up 49 messages. Expression of emotions was 100% and was the highest in this category. Use of humor and self-disclosure were 0% respectively. From the chats 5.3%, 30% and 64.5% constitute affective, cohesive, and interactive indicators of social presence respectively (Fig 1A). Interactive messages have the highest proportion of 64.5% followed by cohesive messages with 30% and lastly affective messages with 5.3%. Furthermore, under the affective component, 100%, 0% and 0% of all messages were categorized as expression of emotions, use of humor and self-disclosure respectively (Fig 1B).

From the result obtained, the group use the WhatsApp platform more for expression of emotions affectively. However, it should be noted that all chats were evaluated and counted for one or more type of level of social presence indicator. Based on interactive response indicator, the group expressed their social presence mostly by engaging in quotes from others' messages (0%), referring explicitly to others' messages (56.6%), asking questions (0.8%), complimenting and expressing appreciation (37%), and expressing agreement (5.5%) (Fig 1C).

The group chat was used more for referring explicitly to other's messages, followed by complementing and expressing appreciation interactively. From the result, vocatives (34.6%), addresses or refers to the group using inclusive pronouns (3.6%), phatic or salutations (61.8%), which are all part cohesive responses (Fig 1D). The overall sum of affective messages is lowest. Table 1: Frequency distribution of chats based social presence indicators.

### Qualitative presentation of findings

Using content analysis of social presence indicators, results were presented in themes [19].

Affective responses–Fig 2 shows the distribution of affective component of social presence indicators. Expression of emotions characteristically involves conventional and

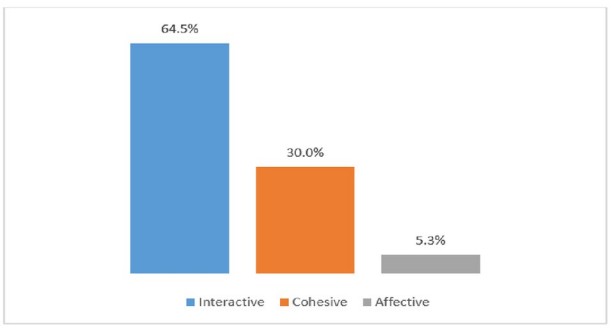

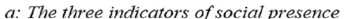

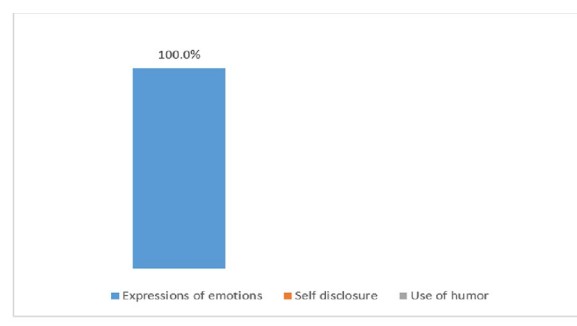

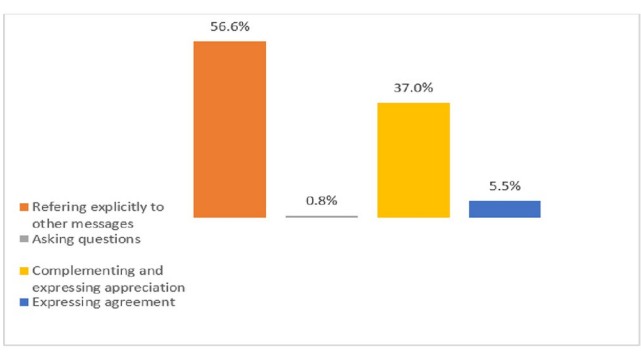

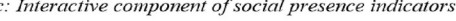

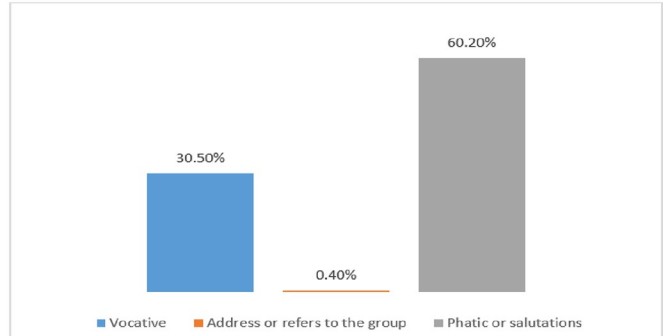

**Fig 1. Frequency distribution of affective, cohesive, and interactive indicators of social presence.** a. The three indicators of social presence. b. Affective component of social presence indicators. c. Interactive component of social presence indicators. d. Cohesive component of social presence indicators.

unconventional use of repetitive punctuations, conspicuous capitalizations and emoticons. For example, the members of the group usually filled the WhatsApp platform with seasonal greetings at festive season and holidays (Appendix 1a in S2 File).

**Table 1. Frequency distribution of chats based social presence indicators.**

| Code | Grounded | Density | Code Group |
|---|---|---|---|
| **Affective** | **1** | **0** | **Affective** |
| Affective (expression of emotion) | 49 | 0 | Affective |
| **Cohesive** | **1** | **3** | **Cohesive** |
| Cohesive | 173 | 3 | Cohesive |
| (Addresses or refer to group Using inclusive pronoun) | | | |
| Cohesive (phatic 'or salutation) | **97** | **2** | **Cohesive** |
| **Interaction** | **1** | **0** | **Interaction** |
| Interaction | 222 | 0 | **Interaction** |
| (Complementing and Expressing appreciation) | | | |
| Interaction | **339** | **0** | **Interaction** |
| (Referring Explicitly to others messages) | | | |
| Interaction (asking question) | **5** | **0** | **Interaction** |
| Interaction | 33 | 0 | **Interaction** |
| **(Expressing agreement)** | | | |

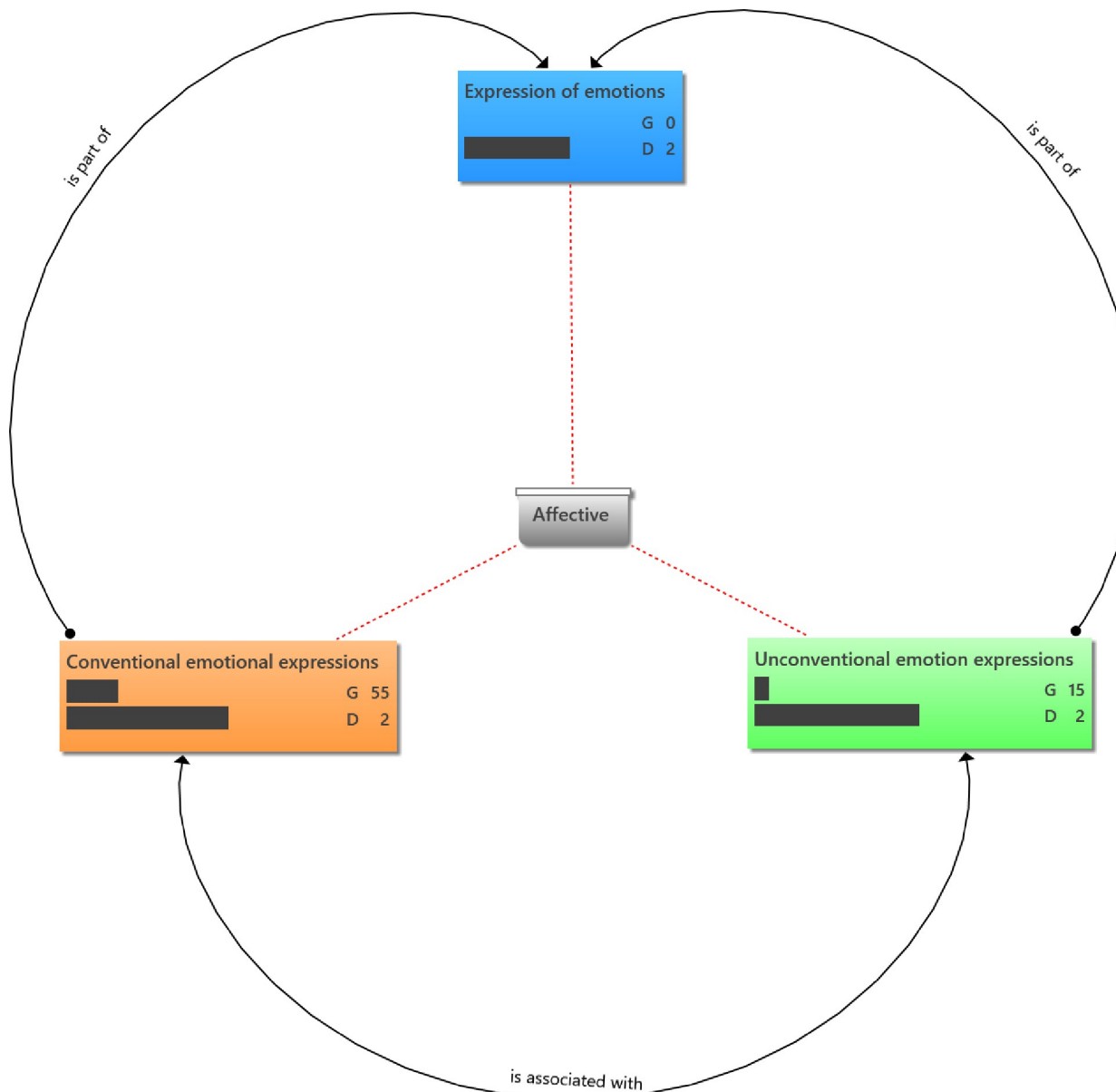

**Fig 2. Distribution of affective components of social presence indicators. Groundedness (G)**–This metric shows the number of quotations linked to a code. Hence it is implied that conventional emotion expressions were more frequent in the conversations than unconventional forms. **Density (D)**–This metric shows the number of codes the stated code is connected to.

Affective (Attention)—Group members, especially its leaders, use the platform as a means to disseminate information that are considered crucial, such as on appointments, meetings, promos, programmes and products, and bereavements (Appendix 1b in S2 File).

Cohesive response–The chats on the WhatsApp of ACAPN are replete with messages suggestive of strong group cohesion. Fig 3 shows the distribution cohesive components of social presence indicators. Members of ACAPN address or refer to group using inclusive pronouns (Appendix 1c in S2 File).

Cohesive (Vocative): In the group chats, there also were many vocative elements where inclusive pronouns such as we, us, and our group, were used (Appendix 1d in S2 File).

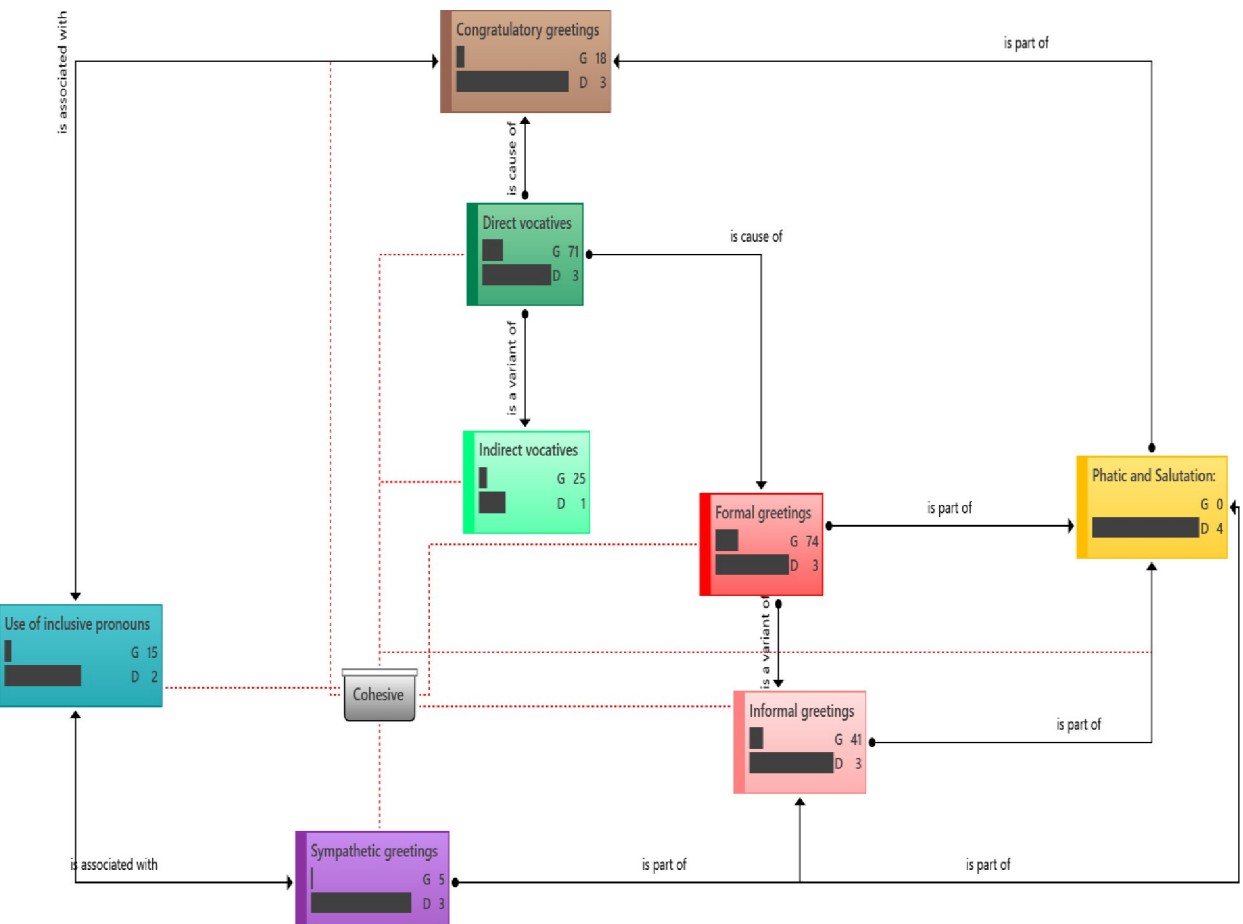

**Fig 3. Distribution of interaction component of social presence indicators. Groundedness (G)**–This metric shows the number of quotations linked to a code. **Density (D)**–This metric shows the number of codes the stated code is connected to. This figure shows the relationship and frequency of different subdivisions in the cohesive category. Formal greetings were found to be more frequent in the conversations than informal greetings with sympathetic greetings being the least frequent.

Cohesive (Phatic or salutation)–Some of the excerpts show the members generously use phatic salutation among themselves as one of the traits of group cohesion (Appendix 1e in S2 File).

**Interaction (complementing and expressing appreciation).** Fig 4 shows the distribution of interaction component of social presence indicators. Among members of the ACAPN, this attribute was commonplace, as seen in some of the excerpts (Appendix 1f in S2 File).

**Interactive (asking question).** Members in the ACAPN group were engaged in asking of questions among themselves as part of the traits of group communication. As such, questions were asked on a myriad of topics. However, most questions were responses/reactions to posts, or to seek clarification on announcements. The WhatsApp group members asked questions that bothered on professional and other matters (Appendix 1g in S2 File).

**Interactive (referring explicitly to others' messages).** Directs reference to content of others is one of the traits of group interactions that was shown on this platform. Members of ACAPN, overwhelmingly respond to post on accomplishments. Some other times, responses were made to post where clarifications were deemed necessary (Appendix 1h in S2 File).

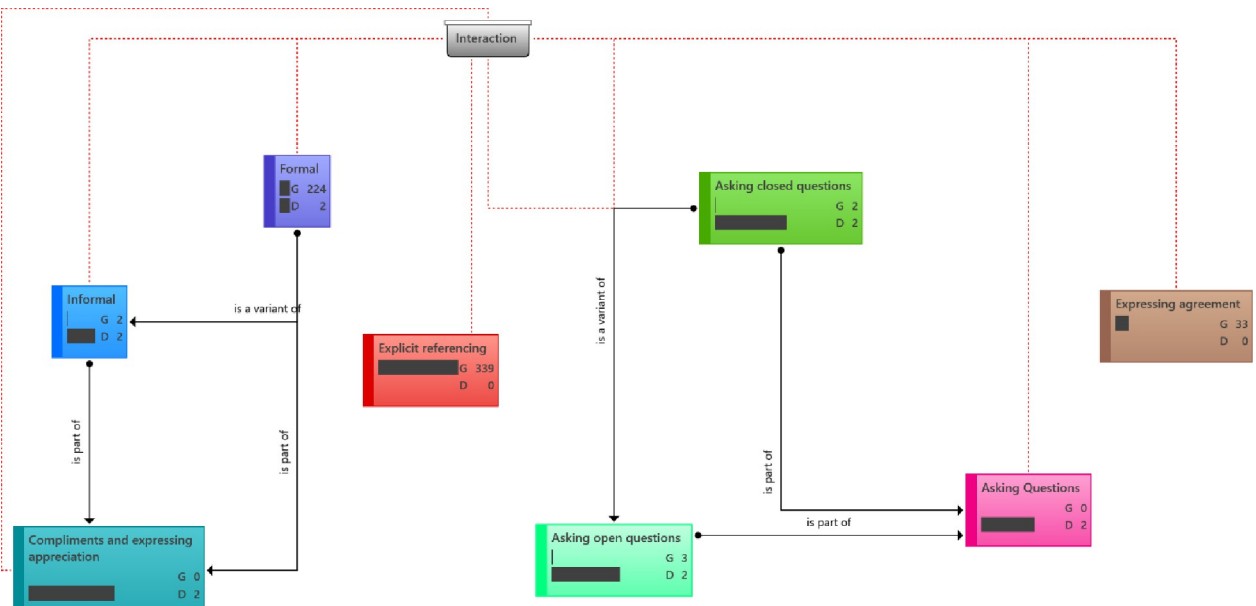

**Fig 4. Distribution of interaction component of social presence indicators. Groundedness (G)**–This metric shows the number of quotations linked to a code. **Density (D)**–This metric shows the number of codes the stated code is connected to. In the interactions observed, formal interactions where significantly frequent than informal forms. Explicit referencing is seen to be the highest in this construct category. Asking of open questions is seen to be slightly higher relatively to closed question.

**Interactive (expressing agreement).** Members of ACAPN express agreement to the messages of others at different context using different words to acknowledge and express agreement to the work of others, to express agreement to the opinions of others, and to commend and express agreement to the work of others (Appendix 1i in S2 File).

## Discussion

In this study, group WhatsApp was found to be a potential medium of communication with peers and promoting social presence among physiotherapists. To our knowledge, this is the first study on this subject among physiotherapists. The use of WhatsApp as a medium to communicate with peers is common in almost in every sector and among professionals. Among healthcare providers, utilization of WhatsApp has been associated with a number of benefits such as improvement in workflows, reduction of phone tags, decrease consultation time, promotion of a collaborative environment to improve the level of healthcare provided to patients, as well as delivery of medical care in a timely and resource-friendly manner [6, 30]. In addition, use of WhatsApp as a group communication platform offers an efficient, unobtrusive and portable mode of communication for medical staff [30]. In this current study, group communication of Nigerian physiotherapists was characterized based on the three social presence indicators of affective, interactive and cohesive responses [21]. From the results, the group chats were mainly composed of interactive messages, followed by cohesive messages, and the least was affective messages. While there is a dearth of specific studies among healthcare professionals with which these findings can be compared, Ohn *et al.* [31] found that undergraduate medical students and lecturer were mostly involved in interactive group communication. In another study among parent groups, interactive trait of social presence was dominant compared with cohesive and affective categories [32].

From the result of this study, most of the WhatsApp chats fell under the category of emotions, which is a sub-code of affective. This indicates that Nigerian physiotherapists used their group WhatsApp platform to express or reflect emotions. According to Moawad *et al.* [33] in a study among general population in Saudi Arabia during the first wave of the coronavirus disease 2019 (COVID-19) pandemic, posts and comments shared on WhatsApp reflected different emotions ranging from negatives to positives. Similarly, Waterloo *et al.* [34] found WhatsApp, followed by Facebook, Twitter, and Instagram as platforms where expressing both negative and positive emotions were perceived appropriate. In line with the above-mentioned, Shahid *et al.* [35] found that group WhatsApp platform was used largely in sharing emoticons and sentiments among students and professionals. Furthermore, the findings from this study show that members of the group used their WhatsApp platform for wishes and greetings. Similar finding was reported in a study by Baishya & Maheshwari [36], which aimed to exploring the academic uses of WhatsApp groups among the students. The authors found that members used the platform to express individual (such as birthday) and group-related (such as festival) wishes. Anecdotally, using group WhatsApp platforms to express wishes and greetings is commonplace and considered appropriate. Some groups will ban posts and comments on topics related to politics and religion, however, general greetings, even on religious occasions are permitted [37].

From this study, leaders and members of the group use the WhatsApp platform to disseminate group-related information that are considered vital. WhatsApp, as an instant messaging app, invented by Brian Acton and Jan Koum in 2009, has been acknowledged as an important social media platform for information sharing and receiving [38], and it is being utilized among professional groups to disseminate information [38–41]. Similar to the observations in this study, professional groups have been found to utilize WhatsApp in sharing messages that can advance social cohesion and socialization [42–44]. Social cohesion among the group in this study was apparent in the WhatsApp chats, as members addressed themselves using first person pronouns such as we, us, and our group/association. It is opined that using 'we' and 'us' is crucial for all members of a group to foster the environment of togetherness [45], and using "we" instead of "I" encourages a sense of collective responsibility and team-mindedness [45, 46]. In addition, from the result of this study greetings and phatic, and questions and answers were employed by members as part of the traits of group communication. Greetings and phatic have been found as part of communicative acts that can improve social presence on online platforms [42, 47]. Also, Baishya & Maheshwari [36] found that the major uses of WhatsApp group among students had to do with questioning to seek or clarity information.

In sum, from the findings of this study, group WhatsApp platform foster group communication and social presence. In particular, the members communicate mostly interactively, affectively and cohesively. These findings are in consonance with reports that WhatsApp group are utilized by professionals for exchanging casual and important information, and keeping in touch with relatives, friends, and colleagues [35, 48–50]. Furthermore, a study by Boulos *et al.* [51] concludes that WhatsApp is effective across a range of social learning and communicative contexts in health and healthcare. However, it is possible that the cohesive leaning of communication on the group WhatsApp in this study was because the physiotherapy association was still in its forming stage of group development (as the group was formed in 2016). According to Tuckman's stages of group development, the forming stage is typically associated with members being polite with one another [52]. Generally, organizations in the early stages of formation exemplify traits of group cohesion, politeness and mutual liking for one another [49].

## Implications and limitations of findings

According to Awada [53], WhatsApp groups helps to enhance interactions, promote dialogue and information sharing, and establish a pleasant atmosphere, as well as advance collaboration with peers. WhatsApp may not be a substitute to short message service (SMS), however, its relatively lower cost, informal nature, ease of coordinating social and working life, and collective discussions on a broad range of topics have given it an edge over the offline platforms of communication [54–59]. Furthermore, the high availability that the app allows, its ease of use, interactive nature and the ability it affords the user to quickly navigate between interpersonal and group communication, have together made WhatsApp an ideal platform for continuous communication [54]. In addition, WhatsApp enables sharing of different-type data and resource, ensuring effective communication and interaction, and even the creating joint activities within created groups [60–62]. On the other hand, WhatsApp being tied to users' mobile phone numbers most often encourage and permits immediate response, which may affect work life balance and one's attention to other matters, as users can also know if their contacts are available online and if message sent has been delivered and read. A potential limitation of this study is the inability to carry out a comparative analysis of the social presence of the group based on social demographic characteristics as a results of anonymization of the data'. It is also possible that the active users on the WhatsApp group may have predominantly shaped the social presence of the group. This is because it was difficult to decipher data from active or passive users, and there were no separate threads or categories for messages, as they all appear in one long string.

## Conclusion

Group WhatsApp platform provided a means of communication and created social presence among Nigerian physiotherapists. Nigerian physiotherapy association communication is mostly interactive, as well as cohesive and affective in terms of dynamics. Thus, WhatsApp communication may be a proxy indicator of level of social presence of group members.

## Supporting information

**S1 File.**
(DOCX)

**S2 File.**
(DOCX)

## Author Contributions

**Conceptualization:** Chidozie E. Mbada, Kikelomo A. Mbada, Clara Fatoye, David Olakorede, Olusola Awoniyi, Udoka A. C. Okafor, Olatomiwa Falade, Francis Fatoye.

**Data curation:** Chidozie E. Mbada, Oluwatosin O. Jeje, Micheal Akande.

**Formal analysis:** Oluwatosin O. Jeje, Micheal Akande, Kikelomo A. Mbada.

**Methodology:** Chidozie E. Mbada, Kikelomo A. Mbada, Clara Fatoye, David Olakorede, Olusola Awoniyi, Udoka A. C. Okafor, Francis Fatoye.

**Software:** Micheal Akande.

**Writing – original draft:** Chidozie E. Mbada, Oluwatosin O. Jeje, Micheal Akande, Kikelomo A. Mbada, Clara Fatoye, David Olakorede, Olusola Awoniyi, Udoka A. C. Okafor, Olatomiwa Falade, Francis Fatoye.

**Writing – review & editing:** Chidozie E. Mbada, Oluwatosin O. Jeje, Micheal Akande, Kikelomo A. Mbada, Clara Fatoye, David Olakorede, Olusola Awoniyi, Udoka A. C. Okafor, Olatomiwa Falade, Francis Fatoye.

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
