## [Decision Letter · Decision Letter 0]

8 Jan 2023

PONE-D-22-30977Social Presence and Dynamics of Group Communication: An Analysis of a Health Professionals WhatsApp Group ChatsPLOS ONE

Dear Dr. ,

Thank you for submitting your manuscript to PLOS ONE. After careful consideration, we feel that it has merit but does not fully meet PLOS ONE’s publication criteria as it currently stands. Therefore, we invite you to submit a revised version of the manuscript that addresses the points raised during the review process.

We look forward to receiving your revised manuscript.

Kind regards,

Latika Gupta

Academic Editor

PLOS ONE

Journal Requirements:

4. Please ensure that you refer to Figures 2, 5, 6 and 7 in your text as, if accepted, production will need this reference to link the reader to the figures.

5. We note you have included a table to which you do not refer in the text of your manuscript. Please ensure that you refer to Table 1 in your text; if accepted, production will need this reference to link the reader to the Table.

Additional Editor Comments:

Please find appended reviewer comments.

Reviewers' comments:

Reviewer's Responses to Questions

**Comments to the Author**

1. Is the manuscript technically sound, and do the data support the conclusions?

Reviewer #1: Yes

Reviewer #2: Partly

2. Has the statistical analysis been performed appropriately and rigorously? 

Reviewer #1: Yes

Reviewer #2: Yes

3. Have the authors made all data underlying the findings in their manuscript fully available?

Reviewer #1: Yes

Reviewer #2: No

4. Is the manuscript presented in an intelligible fashion and written in standard English?

Reviewer #1: No

Reviewer #2: Yes

5. Review Comments to the Author

Reviewer #1: The authors aimed to address Social Presence and Dynamics of Group Communication among Nigerian physiotherapists.

It is a good question and an interesting topic.

However, the article requires some modifications of content and form.

1. Abstract: Please provide keywords in accordance with the authors' guidelines.

2. Methods: insufficient.

Define all the concepts you will develop in the results section (interactive / cohesive / affective com.)

and qualitative concepts (Affective (expression of emotions...)) with examples.

3. Results: must contain the results and only the results.

4. Discussion: Should begin with an answer to the research question.

Then discuss the weaknesses and strengths.

5. The verb tense should be past tense for all sections except the introduction (literature review).

6. All figures should be listed in the corresponding paragraph in the results section.

7. Figures 1, 2, 3, and 4 may be summarized.

8. Table 1, Figures 5, 6, 7 need a legend and footnote to be clearly understood.

9. What does Density mean in the table?

Statistical analysis

It would be relevant to do a comparative analysis based on gender and trend over time.

Can you perform a reproducibility test for your analysis between the 2 admin chats?

Reviewer #2: 1. After harvesting the chat and transferring to Gmail, who owns the data? And whose email ID was used to harvest the same?

2. What about the consent of other physiotherapists in the group, were their members who didn't agree for the same?

3. Qualitative presentation of findings: Would it look better if the examples of various elements in qualitative data are summed up in a separate appendix? This might help in 2 ways: First the look and feel of manuscript may enhance and Second this would help the future authors on the same subject to ease their manuscript without going through the pain of plagiarism correction since the examples given are such commonly used words and sentences.

4. The discussion and implication of findings focuses primarily on uses of WhatsApp with its wide range of utilities in general rather than being specific about the current study. Numerous previous social studies have highlighted what has been presented in the discussion. It would be prudent to review the discussion again to make it more specific to the study actually done.

5. Twice in the manuscript i could see in brackets (Error! Reference source not found), Please clarify the same. The manuscript needs a revision once again to correct few subtle mistakes

6. PLOS authors have the option to publish the peer review history of their article (what does this mean?). If published, this will include your full peer review and any attached files.

Reviewer #1: **Yes: **Ihsane Hmamouchi

Reviewer #2: No

---

## [Author Response · Author response to Decision Letter 0]

14 Feb 2023

PONE-D-22-30977

Social Presence and Dynamics of Group Communication: An Analysis of a Health Professionals WhatsApp Group Chats

Dear Editor,

Thank you for your careful consideration and review of our manuscript. Please find below the point-by-point responses to the comments. 

Editor Comment Response

b) If there are no restrictions, please upload the minimal anonymized data set necessary to replicate your study findings as either Supporting Information files or to a stable, public repository and provide us with the relevant URLs, DOIs, or accession numbers. For a list of acceptable repositories, please see http://journals.plos.org/plosone/s/data-availability#loc-recommended-repositories. This has been done

 4. Please ensure that you refer to Figures 2, 5, 6 and 7 in your text as, if accepted, production will need this reference to link the reader to the figures. This has been done. All figures are referred to. Figures 1-4 are now merged as one.

 5. We note you have included a table to which you do not refer in the text of your manuscript. Please ensure that you refer to Table 1 in your text; if accepted, production will need this reference to link the reader to the Table. This has been done

Reviewer #1 1. Abstract: Please provide keywords in accordance with the authors' guidelines. This has been done

 2. Methods: insufficient.

Define all the concepts you will develop in the results section (interactive / cohesive / affective com.)

and qualitative concepts (Affective (expression of emotions...)) with examples. This has been done 

 3. Results: must contain the results and only the results.

 This has been done 

 4. Discussion: Should begin with an answer to the research question.

Then discuss the weaknesses and strengths. This has been done 

 5. The verb tense should be past tense for all sections except the introduction (literature review). This has been done 

 6. All figures should be listed in the corresponding paragraph in the results section.

 This has been done. 

 7. Figures 1, 2, 3, and 4 may be summarized. This has been done. 

 8. Table 1, Figures 5, 6, 7 need a legend and footnote to be clearly understood. This has been done. 

 9. What does Density mean in the table?

Statistical analysis

It would be relevant to do a comparative analysis based on gender and trend over time.

Can you perform a reproducibility test for your analysis between the 2 admin chats? This was done and reported. The term ‘reproducibility test’ has now been used as appropriate.

Reviewer #2 

 1. After harvesting the chat and transferring to Gmail, who owns the data? And whose email ID was used to harvest the same? Both ethical approval and administrative permission were obtained for this study. The data belongs to the group. OOJ email was used to harvest the chats as text files. 

 2. What about the consent of other physiotherapists in the group, were their members who didn't agree for the same? The researchers made official application for permission to conduct this study to the group. No member of the group disapproved the request. Consequently, an official signed letter to conduct study was issued by the leadership of the group. 

 3. Qualitative presentation of findings: Would it look better if the examples of various elements in qualitative data are summed up in a separate appendix? This might help in 2 ways: First the look and feel of manuscript may enhance and Second this would help the future authors on the same subject to ease their manuscript without going through the pain of plagiarism correction since the examples given are such commonly used words and sentences. An anonymized data (personal identifying information were deleted from the chats (including names, initials, gender, and phone numbers) and were coded appropriately is attached based on PLOS data availability policy. 

The qualitative result section is revised. 

 4. The discussion and implication of findings focuses primarily on uses of WhatsApp with its wide range of utilities in general rather than being specific about the current study. Numerous previous social studies have highlighted what has been presented in the discussion. It would be prudent to review the discussion again to make it more specific to the study actually done. This has been done. 

 5. Twice in the manuscript i could see in brackets (Error! Reference source not found), Please clarify the same. The manuscript needs a revision once again to correct few subtle mistakes This has been done. 

All new inclusion are presented in bold format. Data have been uploaded as supplementary material.

Thank you.

Chidozie Mbada

---

## [Decision Letter · Decision Letter 1]

27 Mar 2023

PONE-D-22-30977R1Social Presence and Dynamics of Group Communication: An Analysis of a Health Professionals WhatsApp Group ChatsPLOS ONE

Dear Dr. Mbada,

Thank you for submitting your manuscript to PLOS ONE. After careful consideration, we feel that it has merit but does not fully meet PLOS ONE’s publication criteria as it currently stands. Therefore, we invite you to submit a revised version of the manuscript that addresses the points raised during the review process.

We look forward to receiving your revised manuscript.

Kind regards,

Latika Gupta

Academic Editor

PLOS ONE

Journal Requirements:

Additional Editor Comments:

Few outstanding comments from a reviewer are appended.

Reviewers' comments:

Reviewer's Responses to Questions

**Comments to the Author**

1. If the authors have adequately addressed your comments raised in a previous round of review and you feel that this manuscript is now acceptable for publication, you may indicate that here to bypass the “Comments to the Author” section, enter your conflict of interest statement in the “Confidential to Editor” section, and submit your "Accept" recommendation.

Reviewer #1: All comments have been addressed

Reviewer #2: (No Response)

2. Is the manuscript technically sound, and do the data support the conclusions?

Reviewer #1: Yes

Reviewer #2: Yes

3. Has the statistical analysis been performed appropriately and rigorously? 

Reviewer #1: Yes

Reviewer #2: Yes

4. Have the authors made all data underlying the findings in their manuscript fully available?

Reviewer #1: Yes

Reviewer #2: Yes

5. Is the manuscript presented in an intelligible fashion and written in standard English?

Reviewer #1: Yes

Reviewer #2: Yes

6. Review Comments to the Author

Reviewer #1: The authors aimed to address Social Presence and Dynamics of Group Communication among Nigerian physiotherapists.

All comments have been adequately addressed.

Reviewer #2: 1. My concern was that examples of various qualitative social presence indicators given in "Results" section should have been provided separately as an appendix or examples should have been given in Material & Methods,after which the examples would have been NOT necessary in Result section, thus refining the results as only and only results and nothing else. For example, In material and methods section, merry Christmas and Happy new year as an example should have been given after defining the Affective response and later the examples are not necessary in results section. Likewise for other parameters.

2. "Error! Reference source not found" is again present in 3rd paragraph of Introduction. Please proofread the manuscript again and delete this sentence.

3. The concern with regard to the Discussion and Implications is yet unresolved. The first third of this section is already highlighted in discussion, thus why repetition? In addition, this section should be Implications and Limitations. Again the

7. PLOS authors have the option to publish the peer review history of their article (what does this mean?). If published, this will include your full peer review and any attached files.

Reviewer #1: **Yes: **Ihsane Hmamouchi

Reviewer #2: No

---

## [Author Response · Author response to Decision Letter 1]

10 May 2023

Social Presence and Dynamics of Group Communication: An Analysis of a Health Professionals WhatsApp Group Chats

Dear Reviewer,

Thank you for your careful consideration and review of our manuscript once again. Please find below the point-by-point responses to the comments. 

Editor Comment Response

Reviewer #2 

 1. My concern was that examples of various qualitative social presence indicators given in "Results" section should have been provided separately as an appendix or examples should have been given in Material & Methods,after which the examples would have been NOT necessary in Result section, thus refining the results as only and only results and nothing else. For example, In material and methods section, merry Christmas and Happy new year as an example should have been given after defining the Affective response and later the examples are not necessary in results section. Likewise for other parameters.

 This has been done. An appendix containing the results is included. Also, the raw data is uploaded as a supplementary material.

 2. "Error! Reference source not found" is again present in 3rd paragraph of Introduction. Please proofread the manuscript again and delete this sentence.

 References in this paragraph were cross-checked. Two of these were replaced for more extant and related ones.

 3. The concern with regard to the Discussion and Implications is yet unresolved. The first third of this section is already highlighted in discussion, thus why repetition? In addition, this section should be Implications and Limitations. Again the The discussion section has been improved upon as suggested. 

All new inclusion are presented in bold format. Data have been uploaded as supplementary material.

Thank you.

Chidozie Mbada

---

## [Decision Letter · Decision Letter 2]

5 Jul 2023

Social Presence and Dynamics of Group Communication: An Analysis of a Health Professionals WhatsApp Group Chats

PONE-D-22-30977R2

Dear Dr. Mbada,

We’re pleased to inform you that your manuscript has been judged scientifically suitable for publication and will be formally accepted for publication once it meets all outstanding technical requirements.

Kind regards,

Nicholas Aderinto Oluwaseyi

Academic Editor

PLOS ONE

Additional Editor Comments (optional):

Reviewers' comments:

Reviewer's Responses to Questions

**Comments to the Author**

1. If the authors have adequately addressed your comments raised in a previous round of review and you feel that this manuscript is now acceptable for publication, you may indicate that here to bypass the “Comments to the Author” section, enter your conflict of interest statement in the “Confidential to Editor” section, and submit your "Accept" recommendation.

Reviewer #1: All comments have been addressed

Reviewer #2: All comments have been addressed

2. Is the manuscript technically sound, and do the data support the conclusions?

Reviewer #1: Yes

Reviewer #2: Yes

3. Has the statistical analysis been performed appropriately and rigorously? 

Reviewer #1: Yes

Reviewer #2: Yes

4. Have the authors made all data underlying the findings in their manuscript fully available?

Reviewer #1: No

Reviewer #2: Yes

5. Is the manuscript presented in an intelligible fashion and written in standard English?

Reviewer #1: Yes

Reviewer #2: Yes

6. Review Comments to the Author

Reviewer #1: Great work! Thank you for taking into account our comments.

Remains few modifications:

References : 13-15 ([Error! Reference source not found.-Error! Reference source not found.) page 3

Table 1: could you add the number of each category ?

Reviewer #2: Dear Authors

Thank you very much for making the changes asked for.

Please proofread the manuscript before final submission.

7. PLOS authors have the option to publish the peer review history of their article (what does this mean?). If published, this will include your full peer review and any attached files.

Reviewer #1: **Yes: **Ihsane Hmamouchi

Reviewer #2: No

---

## [Editor Report · Acceptance letter]

7 Jul 2023

PONE-D-22-30977R2 

Social Presence and Dynamics of Group Communication: An Analysis of a Health Professionals WhatsApp Group Chats 

Dear Dr. Mbada:

I'm pleased to inform you that your manuscript has been deemed suitable for publication in PLOS ONE. Congratulations! Your manuscript is now with our production department. 

Kind regards, 

on behalf of

Dr. Nicholas Aderinto Oluwaseyi 

Academic Editor

PLOS ONE